# Real-World Comparison of Transcatheter Versus Surgical Aortic Valve Replacement in the Era of Current-Generation Devices

**DOI:** 10.3390/jcm12020571

**Published:** 2023-01-10

**Authors:** Young Kyoung Sa, Byung-Hee Hwang, Woo-Baek Chung, Kwan Yong Lee, Jungkuk Lee, Dongwoo Kang, Young-Guk Ko, Cheol Woong Yu, Juhan Kim, Seung-Hyuk Choi, Jang-Whan Bae, In-Ho Chae, Yun-Seok Choi, Chul Soo Park, Ki Dong Yoo, Doo Soo Jeon, Hyo-Soo Kim, Wook-Sung Chung, Kiyuk Chang

**Affiliations:** 1Division of Cardiology, Department of Internal Medicine, Yeouido St. Mary’s Hospital, The Catholic University of Korea, Seoul 07345, Republic of Korea; 2Division of Cardiology, Department of Internal Medicine, Seoul St. Mary’s Hospital, The Catholic University of Korea, Seoul 06591, Republic of Korea; 3Data Science Team, Hanmi Pharm. Co., Ltd., Seoul 05545, Republic of Korea; 4Division of Cardiology, Department of Internal Medicine, Severance Hospital, Yonsei University, Seoul 03722, Republic of Korea; 5Division of Cardiology, Department of Internal Medicine, Korea University Anam Hospital, Korea University, Seoul 02841, Republic of Korea; 6Division of Cardiology, Department of Internal Medicine, Chonnam National University Hospital, Chonnam National University, Gwangju 61469, Republic of Korea; 7Division of Cardiology, Department of Internal Medicine, Samsung Medical Center, Sungkyunkwan University, Seoul 06351, Republic of Korea; 8Division of Cardiology, Department of Internal Medicine, Chungbuk National University Hospital, Chungbuk National University, Cheongju 28644, Republic of Korea; 9Division of Cardiology, Department of Internal Medicine, Seoul National University Bundang Hospital, Seoul National University, Seongnam 13620, Republic of Korea; 10Division of Cardiology, Department of Internal Medicine, St. Vincent Hospital, The Catholic University of Korea, Suwon 65091, Republic of Korea; 11Division of Cardiology, Department of Internal Medicine, Incheon St. Mary’s Hospital, The Catholic University of Korea, Incheon 06591, Republic of Korea; 12Division of Cardiology, Department of Internal Medicine, Seoul National University Hospital, Seoul National University, Seoul 03080, Republic of Korea

**Keywords:** aortic stenosis, transcatheter aortic valve replacement, surgical aortic valve replacement, mortality

## Abstract

Few studies have reported comparisons of out-of-hospital clinical outcomes after transcatheter aortic valve replacement (TAVR) and surgical aortic valve replacement (SAVR) in patients with severe aortic stenosis (AS) in the era of current-generation valves that reflect the real-world situation. Data on patients with severe AS aged 65 years or older who underwent TAVR or SAVR between 2015 and 2018 were obtained from the National Health Insurance Service in Korea and clinical event rate was analyzed. The primary endpoint was all-cause death at 1 year. The cohort included a total of 4623 patients over 65 years of age, of whom 1269 (27.4%) were treated with TAVR. After 1:1 propensity score matching, 2120 patients were included in the study. TAVR was associated with reduced 1-year mortality (hazard ratio (HR): 0.55; 95% confidence interval (CI): 0.42–0.70; *p* < 0.001). There was no difference between the groups in the incidence of ischemic stroke (HR: 0.72, 95% CI: 0.43–1.20; *p* = 0.21) and intracranial hemorrhage (HR: 1.10; *p* = 0.74). Permanent pacemaker insertion was observed more frequently in the TAVR cohort (9.4% vs. 2.5%, HR: 3.95, 95% CI: 2.57–6.09; *p* < 0.001), whereas repeat procedures were rare in both treatments (0.5% vs. 0.3%, *p* = 0.499). In the nation-wide real-world data analysis, TAVR with current-generation devices showed significantly lower 1-year mortality compared to SAVR in severe AS patients.

## 1. Introduction

Transcatheter aortic valve replacement (TAVR) is now accepted as the preferred treatment over surgical aortic valve replacement (SAVR) for very old patients with symptomatic severe aortic stenosis (AS) [1,2]. However, current guidelines for the treatment of severe AS are based on studies examining mostly old-generation TAVR valve devices [3,4,5,6,7,8,9,10,11]. Although there have been studies using newer-generation devices (Evolut Low-Risk, PARTNER 3), these studies have been limited to relatively younger and low-risk patients [12,13]. In addition, for young patients, the current U.S. and European guidelines recommend surgery over transcatheter treatment. Thus, the safety and efficacy of current-generation TAVR valves should be verified in various risk and age groups compared to SAVR in the era of newer devices.

Although TAVR was introduced into global clinical practice more than 20 years ago, it was not introduced in Korea until 2012. Moreover, the number of TAVR cases only began to exceed 100 per year in June 2015, when national insurance began to pay partial costs of TAVR [14]. At present, more than 40 heart centers across the country perform more than 1000 TAVRs annually [15]. At that time of case expansion, current generation TAVR devices, including Sapien 3 (Edwards Lifescience) and Evolut R (Medtronic), have been introduced into Korean TAVR practice, and clinical data and experience have accumulated.

Large-scale randomized controlled trials (RCTs) generate robust statistical evidence and play a pivotal role in evidence-based medicine. However, most large-scale RCTs that compare TAVR- and SAVR-excluded populations with serious illness at the time of screening. Excluding patients with severe comorbidities from RCTs limit the generalizability to real-world clinical practice, especially for fragile geriatric patients who can benefit from less invasive treatment modality. In this regard, analyzing clinical results in real-world clinical practice is important for establishing a strong basis for treatment policies. Furthermore, there is a lack of large-scale clinical data for the relative performance of TAVR versus SAVR in Asian populations. Accordingly, the aim of this study was to compare the clinical outcomes of TAVR and SAVR performed in real-world severe AS patients using the current-generation devices in a broad range of ages.

## 2. Methods

### 2.1. Study Design

The primary endpoint was all-cause death. Ischemic stroke or intracranial hemorrhage (ICH) were analyzed as secondary endpoints. Permanent pacemaker insertion (PPI) and repeat AVR were assessed as safety endpoints.

Data were collected from January 2015 to December 2018 from the National Health Insurance Service (NHIS) program (NHIS-2020-1-177). The NHIS is a mandatory health insurance program provided by the Korean government, covering almost 98% of the Korean population [16]. The NHIS database includes longitudinal follow-up of sociodemographic information, diagnoses, and utilization of healthcare resources, and it has data for each patient. Diagnoses are recorded using International Classification of Diseases–10th Revision (ICD-10) codes, and treatment types are identified using the coding system developed by the Health Insurance Review and Assessment Service (HIRA) [17].

### 2.2. Patients Selection

This is a retrospective cohort study and included patients who received a TAVR or SAVR between January 2015 and December 2018 identified from the Korean NHIS database, and the date of the earliest AVR during the period was set as the index date. To restrict the cohort to patients who received AVR for native valve, those with codes of repeated AVR were excluded from the analysis cohort, and to make study even more comparable, only bioprosthetic valve surgery was included. Additionally, patients who had other indicators of AVR, such as aortic regurgitation (AR), infective endocarditis, and cardiovascular infection (cardiovascular syphilis, gonococcal infection, and candida endocarditis), or underwent surgery with mechanical valve during index hospitalization were excluded. On the contrary, because TAVR is only performed in severe AS patients meeting stringent reimbursement criteria compiled in Korea, no patient who underwent TAVR was excluded on the basis of diagnosis code. Finally, patients younger than 65 years were excluded (Figure 1).

Baseline characteristics were measured to improve the comparability of both treatment groups. Demographic characteristics included age, sex, and clinical characteristics, including comorbidities, such as hypertension, diabetes, heart failure, coronary artery disease (CAD), previous percutaneous coronary intervention (PCI), dyslipidemia, previous stroke, aortic disease, peripheral artery disease (PAD), atrial fibrillation (AF), chronic obstructive pulmonary disease (COPD), chronic kidney disease (CKD), and end-stage renal disease (ESRD).

### 2.3. Statistical Analysis

Descriptive statistics were calculated to show the differences in baseline characteristics of patients in the TAVR and SAVR cohorts. Categorical variables, expressed in percentages or frequencies, were analyzed using chi-square test or Fisher’s exact test as appropriate. Continuous variables, such as age, were expressed as mean ± standard deviation and compared using the unpaired Student’s *t*-test. However, if continuous variable was not normally distributed, it was shown as median and IQR (interquartile range), as well as being compared using Mann–Whitney test. We performed a 1:1 propensity score matching (PSM) for the patients of both groups using a caliper width of 0.25 to reduce selection bias and expressed standardized mean difference. The propensity score was calculated using baseline characteristics, including sex, age, and underlying comorbidities (hypertension, diabetes, heart failure, CAD, previous PCI, dyslipidemia, previous stroke, aortic disease, PAD, AF, COPD, CKD, and ESRD). In the matched cohort, the time to clinical event at 1 month and 1 year were assessed using a Cox proportional hazards model, and the estimates were presented as hazard ratios along with 95% confidence intervals. The log-rank test was performed to assess if there was a significant difference in Kaplan–Meier survival curves between the TAVR and SAVR, and the Schoenfeld test was performed to assess if the hazards were proportional over time. In all comparisons, *p*-values < 0.05 indicated statistical significance. These statistical analyses were performed using the SAS Enterprise Guide version 7.1 (SAS Institute, Cary, NC, USA).

## 3. Results

### 3.1. Baseline Characteristics

Between 2015 and 2018, a total of 10,166 patients who underwent SAVR or TAVR were identified by previously depicted code extracting (Appendix A) from the Korean NHIS database, of whom sixteen patients were excluded because of missing data. After excluding patients with AR and infective endocarditis, with mechanical valve devices, and those under the age of 65, the cohort finally included 4623 patients. Among those, 3354 (72.6%) received SAVR and 1269 (27.4%) underwent TAVR.

Patients in the TAVR group were significantly older than patients in the SAVR group (80.2 ± 5.4 vs. 74.4 ± 5.3, *p* < 0.001; 80, 7 vs. 74, 8, *p* < 0.001). The female patients showed slightly higher trend in the TAVR group (52.7% vs. 49.9%, *p* = 0.092). The proportion of comorbid diseases, except for AF, was also higher in the TAVR group. Baseline demographics and clinical characteristics of selected patients are summarized in Appendix A.

A total of 1060 patients were eligible for 1:1 PSM from the TAVR and SAVR groups, respectively, resulting in balancing the distribution of baseline characteristics between the groups (Table 1 and Appendix A). The mean age was 78.9 ± 4.6 years in the SAVR group and 79.1 ± 4.8 years in TAVR group (*p* = 0.279), and the median ages were 79, 6 (IQR) in both groups (*p* = 0.203). The proportion of female patients was 52.5% in both groups. The proportion of any comorbidities did not differ statistically. The prescriptions of antithrombotics were analyzed and shown in Appendix A.

### 3.2. Procedures

Concomitant coronary revascularization was performed in 12.9% of the patients in the TAVR group and 17.5% of the patients in the SAVR group. Other procedures in the surgery group included valve surgery involving valves other than aortic valve (AV), aortic graft replacement, myectomy, and atrial septal defect closure. In the TAVR group, the transfemoral approach was used in 97.8% of the patients. Balloon-expandable valves were used in 59.7% of the patients with valve systems identified in the TAVR group, and self-expandable valves were used in 37.8%. Details regarding the procedures are provided in Appendix A.

### 3.3. Clinical Outcomes

TAVR showed lower all-cause mortality compared with SAVR at 1 year (8.8% vs. 16.1%, HR: 0.55, 95% confidence interval (CI): 0.42–0.70, *p* < 0.001) (Table 2). The risk of ischemic stroke (2.4% vs. 3.3%; HR: 0.72; 95% CI: 0.43–1.20; *p* = 0.743) and intracranial hemorrhage (2.2% vs. 2.0%, HR: 1.10, 95% CI: 0.61–2.00, *p* = 0.743) were not different between the groups. PPI was performed more frequently in the TAVR group (9.4% vs. 2.5%, HR: 3.95; 95% CI: 2.57–6.09, *p* < 0.001) compared to the surgery group, while the risk of repeat procedure was rare and did not differ between the two groups (0.5% vs. 0.3%; HR: 1.64; 95% CI: 0.39–6.85, *p* = 0.499). The 30-day outcomes are consistent with 1-year outcomes. The 30-day all-cause mortality was lower in the TAVR group (3.2% vs. 7.8%; HR: 0.40; 95% CI: 0.27–0.60; *p* < 0.001), and permanent pacemaker implantation occurred frequently in patients who underwent TAVR (8.2% vs. 1.3%; HR: 6.28; 95% CI: 3.57–11.0, *p* < 0.001). The Kaplan–Meier estimates of each clinical event are shown in Figure 2. In the subgroup analysis, there was no significant difference in all-cause mortality according to the underlying diseases or demographic factors (Figure 3).

## 4. Discussion

The major finding of this study is that 1-year all-cause death was significantly lower in patients aged 65 years or older undergoing TAVR for severe AS than with SAVR. TAVR showed significant differences in 30-day mortality at the beginning of the procedure and maintained this trend for up to one year. Among large-scale RCTs, only the CoreValve High Risk trial demonstrated that TAVR significantly reduced the all-cause death compared with SAVR at one-year outcomes, where old-generation self-expandable valves were used [6].

Several large registries have not yet reported the benefit of the TAVR when extending to out-of-hospital follow-up. The reports of most registries to date show that mid-term performance of TAVR is comparable or inferior to surgery. Data from the GARY (German Aortic Valve Registry), the world’s largest TAVR registry, indicate one-year results for TAVR and SAVR were similar, while five-year results by PSM showed an inferior outcome of TAVR [18]. The OBSERVANT study (Observational Study of Effectiveness of SAVR-TAVI Procedures for Severe Aortic Stenosis Treatment) also reported that SAVR was superior to transfemoral TAVR using early-generation devices at 5-year follow-up [19]. Likewise, a U.S. registry showed similar 1-year outcomes [20]. However, it should be noted that those studies were based on old-generation valves.

Our findings may be the first to demonstrate superiority of TAVR over SAVR in reducing mortality at one-year follow-up in a nation-wide observational data. The superiority of TAVR in this study is believed to be due to several factors. First, this study enrolled patients who underwent TAVR using mostly new-generation valves, while previous studies recruited patients treated with old generation valves. If procedural skill improvement, the so called “learning curve effect”, dedicated patient selection, and periprocedural management have a great influence on in-hospital outcomes, more sophisticated contemporary valve systems are also thought to contribute to out-of-hospital outcome improvement [18,21,22]. In fact, a Korean study comparing clinical outcomes between Sapien XT and Sapien 3 showed significantly lower 1-year cardiovascular mortality in patients treated with Sapien 3 [23]. Second, this is the first large-scale study of Asian populations, while pivotal RCTs and large-scale retrospective studies were mostly conducted in patients from North America and Western Europe. It should be considered that the small habitus of Asian are disadvantageous to invasive treatments, such as open heart surgery. Smaller patients are forced to be inserted with small valves, have patient-prosthesis mismatch (PPM), and worsen clinical outcomes. Considering information from previous studies that show transcatheter procedures tend to use larger valves compared to surgery, this benefit can be maximized in Asian people. Third, in Korea, the introduction of TAVR was later than in the other developed countries, which resulted in the procedure being performed after the knowledge of the advantages and disadvantages of certain valves according to the patient’s profile, such as aortic root anatomy, being established. In the actual practice, we confront different settings unlike in pivotal RCTs because a valve type suitable for the patient’s anatomy is selected instead of the fixed assignment of a valve system. In that respect, the study that resembles the present study is UK TAVI (UK Transcatheter Aortic Valve Implantation) [24]. In contrast to previous pivotal RCTs, UK TAVI was designed to compare TAVR and surgery using any valve type in a broad range of patients. That study did not show statistical survival difference between treatment groups; however, the transcatheter group showed a numerically lower trend of mortality.

In this study, the surgery group showed numerically higher mortality than that observed in previous studies [25,26]. We assumed that PSM might eliminate the low-risk patients and assign more elderly patients with poor comorbidities to the SAVR cohort than crude analyses. In addition, the study cohort excluded patients under 65, and concomitant procedures may have contributed to a decrease in the survival rate of the surgery group (Appendix A). Furthermore, it seems that the small build of the elderly in Korea, who went through the era of underdeveloped countries as a growing period, also worked against the outcomes of surgery.

Although large-scale RCTs for patients at various surgical risk levels demonstrated acceptable efficacy and safety outcomes with TAVR, TAVR has not yet been established as a standard treatment for relatively younger patients. This is due to the lack of long-term evidence of efficacy and safety of TAVR compared to SAVR [27,28]. Even the CoreValve High Risk trial, the only randomized trial to demonstrate superior 1-year survival with TAVR, showed no difference at 5 years of follow-up.

This study provides new information about TAVR. First, the initial survival gain of the transcatheter procedure has sustained to out-of-hospital follow-up until one year by Kaplan–Meier analysis, suggesting a rationale for prioritizing TAVR in the treatment of severe AS. Second, TAVR can be a more beneficial treatment in Asian people. Third, there were no significant interactions between treatment modality and underlying co-morbidities that had been excluded from major RCTs, such as dialysis. Fourth, as is well known, TAVR was associated with a higher incidence of PPI. However, the absolute incidence of PPI in this study was relatively low compared to previous reports, even more so considering that the TAVR group contained about a third of patients with self-expandable valves [29,30]. It is necessary to study what contributed to these results.

This study has several limitations. First, the dataset analyzed was from the health insurance system in which researchers use codes to include or exclude patients and analyze clinical events. However, incorrect entry of disease codes is a serious shortcoming of the insurance data-based study. To minimize these shortcomings, we used claims codes and death events that are unlikely to differ from the actual practice and situation. Claims codes are heavily scrutinized and the information on survival or death is accurately monitored by the HIRA in Korea because it is related to financial reimbursement. Whereas many studies based on insurance data reported to date are dependent on disease codes, this study has a unique advantage in that it used claims codes. Second, ideal comparison of both treatment can be performed under the setting of isolated AVR. However, concomitant surgery, such as coronary artery bypass graft, valve surgery not for AV made the SAVR cohort unfavorable compared to TAVR, where PCI was a mainstay of concurrent procedures. Moreover, the fact that the frequency of coronary revascularization was higher in the surgical group also adversely affected the SAVR (*p* = 0.004). On the contrary, repeat AVR was excluded in SAVR; however, this was not the case in TAVR, which was disadvantageous to TAVR. Additionally, since concomitant procedures were collected during the index hospitalization period, it should be considered that there are cases that were not performed simultaneously on the same date. Third, valve size profile, key biochemical and imaging data for paravalvular leakage, PPM, and structural valve degeneration or bicuspid etiology were unavailable [31,32,33,34,35,36,37]. In addition, perioperative risk scoring systems, such as the STS-PROM (Society of Thoracic Surgeons-Predicted Risk of Mortality) and EuroScore II, which are widely used in the clinical practice and research field of valve treatment, could not be determined from the data. Fourth, the cause of death was not accurately determined due to the limitations of information. Therefore, this study analyzed all-cause mortality, not cardiovascular death. Nonetheless, it is encouraging that even the all-cause death is significantly lower in the TAVR group compared with those undergoing SAVR. Fifth, 1-year follow-up in this study was relatively short. It was not possible to set a longer follow-up period to completely obtain the clinical results, owing to the truncation of the data set. In the five-year outcomes of the PARTNER 2 trial, TAVR with the SAPIEN-XT, the previous-generation valve, showed a higher incidence of all-cause death over time [27]. Longer follow-up examining outcomes with current generation valves is needed to draw a firm conclusion about long-term outcomes following TAVR. Sixth, this study is conducted on patients from Korea. There may be limitations to generalizing the results to other populations [38,39,40]. Finally, there is concern that patients in good condition are inevitably assigned to surgery and poor condition to less invasive treatment in observation studies, which is an insurmountable hurdle of statistical technique. However, this study cohort overcame these handicaps and demonstrated superior survival of less invasive treatment by retrospective analysis.

In spite of the aforementioned limitations, current-generation device using TAVR significantly reduced all-cause death compared with SAVR in severe AS patients over 65 years old at 1-year follow-up from this nation-wide real-world analysis. A higher rate of permanent pacemaker insertion is associated with TAVR, but the absolute rate was reasonable.

## 5. Conclusions

In real-world retrospective data, TAVR showed similar or inferior mid-term outcomes compared with SAVR in various nation-wide studies. However, retrospective studies comparing TAVR with SAVR in patients with severe AS were mostly performed with old-generation valve systems. In particular, large-scale data of Asian people for comparison of TAVR versus SAVR were scarce. In this Korean nation-wide real-world data analysis, TAVR with current-generation devices showed lower mortality after mid-term follow-up compared to surgery in severe AS patients, despite the higher PPI rates.

## Figures and Tables

**Figure 1 jcm-12-00571-f001:**
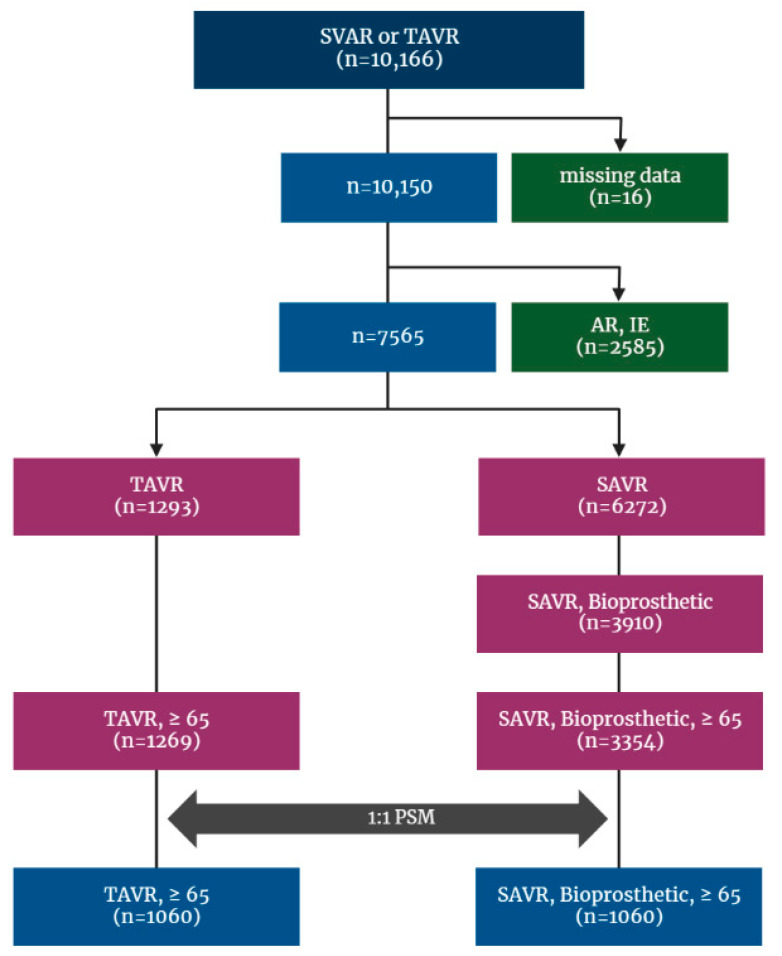
Flowchart for patient enrollment and exclusion. AR: aortic regurgitation; AS: aortic stenosis; IE: infective endocarditis; NHIS: National Health Insurance Service; PSM: propensity score matching; SAVR: surgical aortic valve replacement; TAVR: transcatheter aortic valve replacement.

**Figure 2 jcm-12-00571-f002:**
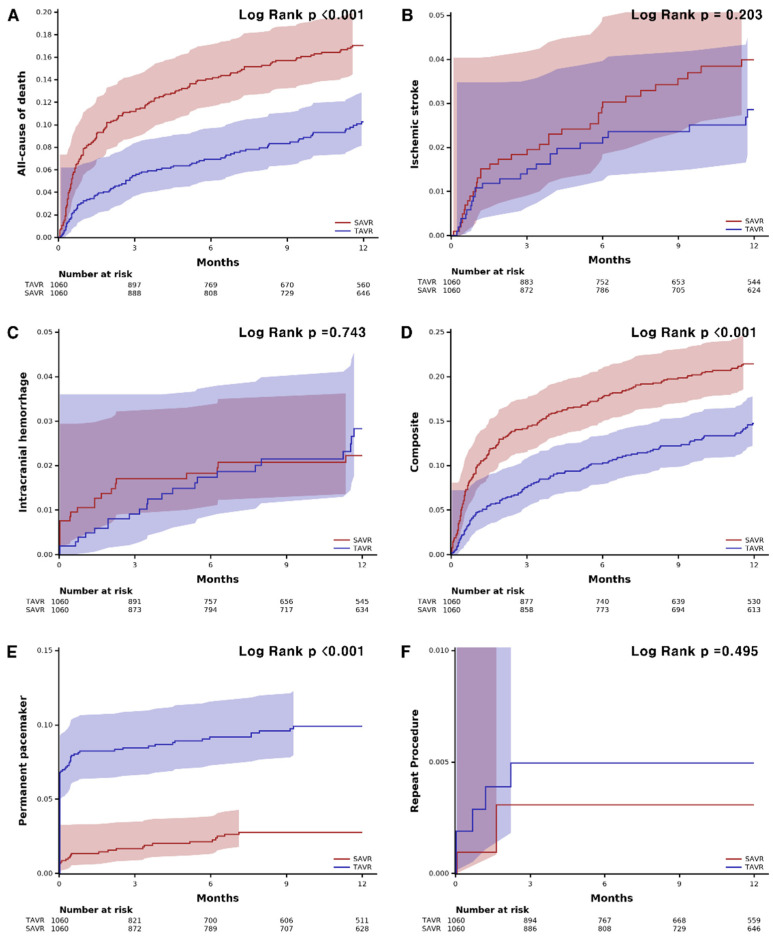
Kaplan–Meier curves of clinical events and safety outcomes. (**A**) all-cause death. (**B**) ischemic stroke. (**C**) intracranial hemorrhage. (**D**) composite of all-cause death, ischemic stroke, or intracranial hemorrhage. (**E**) permanent pacemaker. (**F**) repeat procedure. SAVR, surgical aortic valve replacement; TAVR, transcatheter aortic valve replacement.

**Figure 3 jcm-12-00571-f003:**
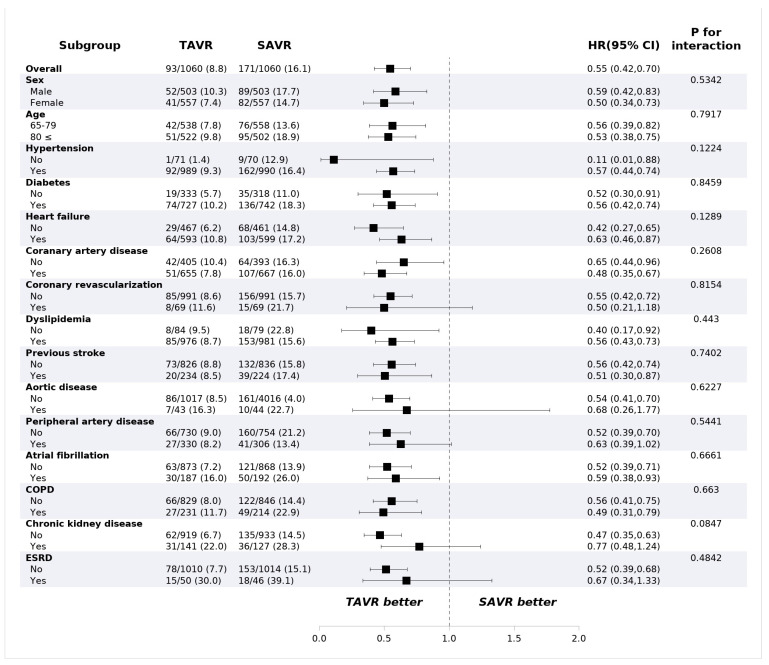
Mortality of subgroups in propensity-score-matched cohorts. Values are presented as number (%). CI: confidence interval; COPD: chronic obstructive pulmonary disease; ESRD: end-stage renal disease; HR: hazard ratio; SAVR: surgical aortic valve replacement; TAVR: transcatheter aortic valve replacement.

**Table 1 jcm-12-00571-t001:** Baseline characteristics of patients after propensity score matching.

	TAVR *n* = 1060	SAVR *n* = 1060	SMD	*p*-Value
Age (mean ± SD)	79.1 ± 4.8	78.9 ± 4.6		0.279
(median, IQR) *	79, 6	79, 6	0.047	0.203
Age categories			−0.019	1.000
65~79	538 (50.8)	568 (53.4)		
≥80	522 (49.2)	502 (47.4)		
Female	557 (52.5)	557 (52.5)	0	1.000
Hypertension	989 (93.3)	990 (93.4)	0.001	0.931
Diabetes	727 (68.6)	742 (70.0)	0.014	0.480
COPD	231 (21.8)	214 (20.2)	−0.016	0.365
CAD	655 (61.8)	667 (62.9)	0.011	0.591
Previous PCI	69 (6.5)	69 (6.5)	0	1.000
Dyslipidemia	976 (92.1)	981 (92.5)	0.005	0.684
Heart failure	593 (55.9)	599 (56.5)	0.014	0.793
AF	187 (17.6)	192 (18.1)	0.005	0.777
Previous stroke	234 (22.1)	224 (21.1)	−0.009	0.598
Aortic disease	43 (4.1)	44 (4.2)	0.009	0.913
PAD	330 (31.1)	306 (28.9)	−0.023	0.255
CKD	141 (13.3)	127 (12.0)	−0.013	0.360
ESRD	50 (4.7)	46 (4.3)	−0.004	0.676

AF, atrial fibrillation; CAD: coronary artery disease; CKD: chronic kidney disease; COPD: chronic obstructive pulmonary disease; ESRD: end-stage renal disease; IQR; interquartile range; PAD: peripheral artery disease; PCI: percutaneous coronary intervention; SAVR: surgical aortic valve replacement; SD: standard deviation; SMD: standardized mean difference; TAVR: transcatheter aortic valve replacement.; * Mann–Whitney test was additionally performed because age variable did not show normality of distribution.

**Table 2 jcm-12-00571-t002:** Clinical events and safety outcomes in propensity-score-matched cohorts.

	At 1 Month					At 1 Year				
	TAVR *n* = 1060	SAVR *n* = 1060	HR	95% CI	*p*-Value	TAVR *n* = 1060	SAVR *n* = 1060	HR	95% CI	*p*-Value
All-cause death	34 (3.2)	83 (7.8)	0.40	0.27–0.60	<0.001	93 (8.8)	171 (16.1)	0.55	0.42–0.70	<0.001
Ischemic stroke	11 (1.0)	11 (1.0)	0.98	0.42–2.26	0.958	25 (2.4)	35 (3.3)	0.72	0.43–1.20	0.743
ICH	4 (0.4)	11 (1.0)	0.36	0.12–1.13	0.080	23 (2.2)	21 (2.0)	1.10	0.61–2.00	0.743
Repeat AVR	3 (0.3)	1 (0.1)	2.98	0.31–28.65	0.344	5 (0.5)	3 (0.4)	1.64	0.39–6.85	0.499
PPI	87 (8.2)	14 (1.3)	6.28	3.57–11.0	<0.001	100 (9.4)	26 (2.5)	3.95	2.57–6.09	<0.001

Event is denoted as number (%). Hazard ratio was assessed by Cox proportional hazard model. AVR: surgical aortic valve replacement; CI: confidence interval; HR: hazard ratio; ICH: intracranial hemorrhage; PPI: permanent pacemaker insertion; SAVR: surgical aortic valve replacement; TAVR: transcatheter aortic valve replacement.

## Data Availability

The crude data presented in this study are available under permission by Korean National Health Insurance.

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
