# Peer review of "Real-World Comparison of Transcatheter Versus Surgical Aortic Valve Replacement in the Era of Current-Generation Devices"

_jcm, 2023, doi:10.3390/jcm12020571_

Round 1

Reviewer 1 Report

 Kyoung Sa et al reported their work entitled “Real-World Comparison of Transcatheter Versus Surgical Aor-2 tic Valve Replacement in the Era of Current-Generation De vices” and concluded that “In the nation-wide real-world data analysis, TAVR with current-generation devices showed significantly lower 1-year mortality compared to SAVR in severe AS patients.”.

-          For PSM analysis, please report standardized mean difference and love plot rather than reporting P value. SMD of 0.10 or less reflects proper matching. Please edit the current table 1 and amend your method section.

-          Please test continuous variables for normality and report median and interquartile range rather than mean and SD if variables were not normally distributed and use Mann Whitney test instead of t test.

-          Please add 95% confidence intervals to your Kaplan Meier curves in Figure 2.

-          This is one of the latest meta-analysis that could be cited “https://pubmed.ncbi.nlm.nih.gov/35216819/”

Author Response

Thank very much for reviewing the article.  We authors are grateful your appreciation for our research that compare clinical outcome of transcatheter procedure and surgery in severe AS patients. I hope this study will be published in the Journal of Clinical Medicine so that Asian TAVR data can be shared globally.

Point 1.

We have re-established PSM method according to your directions, and marked standardized mean differences. Table 1 and paragraph in Method section have been revised.

Point 2.

The statistical analysis was modified as recommended for the non-normally distributed continuous variables.

Point 3.

As recommended, 95% confidence intervals were added to the curve.

Point 4.

Thank for your citation recommendation. We think that the knowledge and information of the article is enriched by adding a recent meta-analysis paper.

Kind regards,

Kiyuk Chang

kiyuk@catholic.ac.kr

Reviewer 2 Report

Dear authors:

Excellent. I am impressed. First of all, I congratulate you on your national database, which is lacking in so many countries that otherwise claim to be at the forefront of medical innovation. I very much appreciate your findings and agree with them wholeheartedly. The time of surgical valves is over, i.e. they are reserved for cases that are not candidates for primary TAVR. Allow me to ask for some adjustments to your manuscript:

1. could you provide data on the outcomes of Sapien valves versus Corevalve/Evolut valves, i.e. self-expanding valves versus balloon-expandable valves in terms of the need for a pacemaker? This would be of interest to readers and to insurance companies who generally pay more for Sapien valves than Corevalve or Evolut valves. The presumed higher pacing requirement might indeed offset the initial saving of using self-expanding valves.

2. can you show the differences in outcomes between mechanical and biological valves?

3. since not so many mechanical valves were implanted overall, could you produce the same statistics with biological surgical valves only? That would make it even more comparable, while on the other hand you would not lose so many cases in a propensity score analysis. 

Author Response

Thank very much for reviewing the article.  We authors are grateful your appreciation for our research that compare clinical outcome of transcatheter procedure and surgery in severe AS patients. I hope this study will be published in the Journal of Clinical Medicine so that Asian TAVR data can be shared globally.

Point 1.

Since the subject of this study is to compare the clinical result of surgery and transcatheter procedure, the comparison of balloon-expandable valve and self-expandable valve was an analysis that was not originally thought of in the study setting. We are also interested in the clinical result of the two TAVR valve systems, so we would like to conduct a separate study, and it is regretful that we cannot provide data together. We are sorry and hope your mercy.

Point 2.

Since this dataset is limited to patients aged 65 or older, the number of patient treated by mechanical valve is only 31 out of 1,063 surgery patients, so it is considered difficult to compare the tissue and mechanical valve in this study.

Point 3. 

We also think it is reasonable to limit valve type of the surgical group to tissue valve. We have re-established the study subject by actively reflecting your recommendation. The result were consistent with previous analysis.

Kind regards,

Kiyuk Chang

kiyuk@catholic.ac.kr

Reviewer 3 Report

This is a retrospective analysis of a large scale database comparing the clinical outcomes of TAVR and SAVR performed in real-world severe AS patients using the current-generation devices in a broad range of ages. The major finding of this study is that 1-year all-cause death was significantly lower in patients aged 65 years or older undergoing TAVR for severe AS than with SAVR. TAVR showed significant differences in 30-day mortality at the beginning of the procedure and maintained this trend for up to one year.

Author Response

Thank very much for reviewing the article.  We authors are grateful your appreciation for our research that compare clinical outcome of transcatheter procedure and surgery in severe AS patients. I hope this study will be published in the Journal of Clinical Medicine so that Asian TAVR data can be shared globally.

Kind regards,

Kiyuk Chang

kiyuk@catholic.ac.kr

Round 2

Reviewer 1 Report

Dear authors

Thanks for your edits and it is my pleasure to accept your work.

Minor post-acceptance edits:

- Please delete the old Figure 2 (Kaplan Meier curves) without confidence intervals.

Author Response

It is understood that your meaning is to remove previous Figure 2 without confidence interval on it, but I have already uploaded a revised version with previous Figure 2 that you point out erased. I wonder if it was intended to make there no confidence interval in the final Figure 2.

Reviewer 2 Report

Dear authors: I appreciate your thorough review. Especially that you took the trouble to re-compare the statistics after eliminating the mechanical valves. The manuscript now has a clear message. Thank you.

Author Response

Thank you for reviewing our article. It is honor to introduce Asian clinical data about comparison between TAVR and SAVR in Journal of Clinical Medicine. We authors wish Happy New Year to you.